# Hybrid-Flash Butterfly Optimization Algorithm with Logistic Mapping for Solving the Engineering Constrained Optimization Problems

**DOI:** 10.3390/e24040525

**Published:** 2022-04-08

**Authors:** Mengjian Zhang, Deguang Wang, Jing Yang

**Affiliations:** 1Electrical Engineering College, Guizhou University, Guiyang 550025, China; 202111088258@mail.scut.edu.cn (M.Z.); dgwang@gzu.edu.cn (D.W.); 2School of Computer Science and Engineering, South China University of Technology, Guangzhou 510006, China

**Keywords:** hybrid-flash butterfly optimization algorithm, firefly algorithm, logistic mapping, optimization, engineering constrained issues

## Abstract

Only the smell perception rule is considered in the butterfly optimization algorithm (BOA), which is prone to falling into a local optimum. Compared with the original BOA, an extra operator, i.e., color perception rule, is incorporated into the proposed hybrid-flash butterfly optimization algorithm (HFBOA), which makes it more in line with the actual foraging characteristics of butterflies in nature. Besides, updating the strategy of the control parameters by the logistic mapping is used in the HFBOA for enhancing the global optimal ability. The performance of the proposed method was verified by twelve benchmark functions, where the comparison experiment results show that the HFBOA converges quicker and has better stability for numerical optimization problems, which are compared with six state-of-the-art optimization methods. Additionally, the proposed HFBOA is successfully applied to six engineering constrained optimization problems (i.e., tubular column design, tension/compression spring design, cantilever beam design, etc.). The simulation results reveal that the proposed approach demonstrates superior performance in solving complex real-world engineering constrained tasks.

## 1. Introduction

Many meta-heuristic algorithms have been proposed and successfully used for solving numerical optimization problems [1], engineering design problems [2], future selection [3], etc. For the swarm intelligence algorithm, the classical method is particle swarm optimization (PSO) algorithm [4], which is widely used in various fields [5]. There are many novel approaches proposed in recent years, which are inspired by the group behavior of the animals in nature, such as ants, bees, fireflies, wolves, and so on. These swarm algorithms are named like ant colony optimization (ACO) [6], bee colony optimization (BCO) algorithm [7], firefly algorithm (FA) [8], grey wolf optimizer (GWO) [9], etc. Moreover, the classical evolution algorithms, such as genetic algorithm (GA) [10] and differential evolution (DE) [11], are well-known in the evolution computation field.

A recent introduction in the swarm intelligence field is butterfly optimization algorithm (BOA) [12], which imitates the foraging and mating behavior of the butterfly in nature. To improve the global ability of the basic BOA, there are various improved versions of BOA that have been developed, which can be divided into the improvement of control parameters [13,14] and hybrid algorithm [15] from the improved strategies in brief. Additionally, the major applications of BOA and its variants are, including but not limited to, the node localization in wireless sensor networks [16], optimization of wavelet neural networks [17], and the feature selection task [18]. For more details, Zhang et al. [15] proposed a hybrid algorithm for solving the high-dimensional numerical optimization tasks, named HPSOBOA, which is combined BOA with PSO, and cubic mapping is used to adjust the parameter *a*. Moreover, EL-Hasnony et al. [19] improved this method to solve the feature selection task, which includes a case dataset of COVID-19. In addition, An et al. [20] used the HPSOBOA method to solve the inverse kinematic problem. However, no available information in the above studies was presented for modifying the BOA’s mathematical model to match the actual foraging behavior of butterflies.

A novel hybrid-flash butterfly optimization algorithm (HFBOA) is proposed for solving the constrained engineering problems, aiming to improve the optimization accuracy of the original BOA. Only the smell perception rule in foraging and mating is considered, which leads to the poor optimal precision of the BOA. Relevant ecological studies [21] have shown that the vision of butterflies plays a critical role in searching for food (collecting pollen). In the HFBOA, smell and vision are taken into consideration for global and local search, respectively, which makes it more in line with the actual butterfly foraging characteristics. Furthermore, updating the control parameters by logistic mapping is used to enhance the global optimization capability of the HFBOA. According to the no free lunch (NFL) theory [22], a new optimization algorithm should be used to deal with a certain type of problem.

To verify the performance of the proposed algorithm, 12 benchmark test functions are fairly selected from the CEC benchmark functions in this paper. The performance of the HFBOA was compared with six state-of-the-art meta-heuristic methods. The comparison results show that the proposed algorithm is not only superior to the original BOA but also its variants, such as LBOA [16], IBOA [13], MBOA [14], HPSOBOA [15], and other meta-heuristic algorithms, which indicates the effectiveness of HFBOA. Furthermore, HFBOA has been successfully used to solve six constrained engineering problems, which are tubular column design, three bar truss design, tension/compression spring design, welded beam design, cantilever beam design, and speed reducer design optimization problems. In general, the main highlights and contributions of the proposed method are summarized as follows: (i) A novel hybrid-flash butterfly optimization algorithm (HFBOA) is proposed. (ii) HFBOA has an overall competitive performance according to the statistical results and convergence curves. (iii) We also used the proposed approach to solve six constrained engineering problems, which are compared with many advanced methods.

The rest of this paper is organized as follows: Section 2 presents the mathematical model of the original BOA. In Section 3, a novel hybrid-flash butterfly optimization algorithm is proposed, and chaos improvement strategies and time complexity of HFBOA are also presented. Section 4 illustrates comparative analysis for solving the numerical optimization and engineering constrained optimization problems, the experimental results are also performed in detail. Section 5 presents the discussion of the proposed method for solving the numerical and engineering optimization issues. Finally, the conclusions and future studies are summarized in Section 6.

## 2. Model of the Basic BOA

The BOA [12] is based on the foraging and mating behavior of the butterfly in nature. Three phases (fragrance, global search, local search) are presented in the basic BOA. The fragrance is given by:(1)$$f_{i}={cI}^{a}$$
where $f_{i}$ is the perceived magnitude of fragrance, *c* represents the sensory modality, *I* is the stimulus intensity, and *a* represents the power exponent based on the degree of fragrance absorption.

The sensory modality *c* in the optimal search phase of the basic BOA is given by:(2)$$c_{t+1}=c_{t}+\frac{0.025}{c_{t}}\cdot T_{max}$$
where $T_{max}$ is the maximum number of evolutionary iterations, and the initial value of parameter *c* is set to 0.01. According to Equation (Equation 2), the value range of the parameter *c* is in (0, 1, 0.3).

A switch parameter *p* in (0, 1) is used to choose the pase between global search and local search. The global search movement of the butterfly is given by:(3)$$x_{i}^{t+1}=x_{i}^{t}+\left(r^{2}\times g_{best}-x_{i}\right)\times f_{i}$$
where $x_{i}^{t}$ denotes the solution vector $x_{i}$ of the *i*th butterfly in the *t* iteration and *r* is a random number in [0, 1]. Here, $g_{best}$ is the current best solution found among all each stage solutions. The local search phase is given by:(4)$$x_{i}^{t+1}=x_{i}^{t}+\left(r^{2}\times x_{i}^{k}-x_{j}^{t}\right)\times f_{i}$$
where $x_{j}^{t}$ and $x_{i}^{k}$ are the *j*-th and *k*-th butterflies chosen randomly from the solution space. If $x_{j}^{t}$ and $x_{i}^{k}$ belong to the same iteration, it means that the butterfly becomes a local random walk. If not, this kind of random movement will diversify the solution.

## 3. Hybrid-Flash Butterfly Optimization Algorithm

We have taken the search strategy of the firefly algorithm (FA) [8] into consideration, and used vision of butterflies for the local optimization in HFBOA. Each stage of HFBOA is presented, including the initialization phase, optimization phase, global search, local search, and switch parameter setting.

### 3.1. Initialization Phase

The population initialization positions of the butterflies are set by the random function, here, the general formula of the initial position is as follows:(5)$$X_{i,j}=X_{lb,j}+rand\times \left(X_{ub,j}-X_{lb,j}\right)$$
where $x_{i,j}$ is the *i*-th solution for the *j*-th dimension, i∈[1,2,3,⋯,n], j∈[1,2,3,⋯,Dim]. $x_{ub,j}$ and $x_{lb,j}$ represent the upper and lower bounds of the problem, respectively, and rand is a uniform random number in [0, 1]. This strategy is usually used to initialize the position of the population of the swarm intelligence algorithms.

### 3.2. Optimization Phase

Particularly, $F_{i}^{t}$ represents the fragrance of the *i*-th butterfly in the *t*-th iteration, it can be calculated as:(6)$$F_{i}^{t+1}=c\cdot {\left(F_{i}^{t}\right)}^{a}$$
where *c* is the sensory modality, it can be set to a random number in (0, 1) during the search stage of the HFBOA. Due to the interval (0, 1) of parameter *c*, we use the chaotic strategy to update its value with a one-dimensional chaotic mapping, named logistic mapping. In the original BOA, the power exponent *a* is set to 0.1, thus, we also take it in the proposed method in the following experiments.

### 3.3. Global Search

We take the parameter *r* into consideration so that α is used to replace it. Hence, the mathematical model of the butterflies’ global search movements of the proposed approach can be formulated as follows:(7)$$X_{i}^{t+1}=X_{i}^{t}+\left(\alpha^{2}\times g_{best}-X_{i}\right)\times F_{i}^{t}$$
where $X_{i}^{t}$ represents the solution vector $X_{i}$ of the *i*-th butterfly in the *t*-th iteration and α is a random number in (0, 1). $g_{best}$ is the current best solution found among all the solutions in the current stage. To some extent, parameter α can be regarded as a scaling factor, which is utilized to adjust the distance between the *i*-th butterfly and the best solution.

### 3.4. Local Search

Two phases of the HFBOA should be switched when the individuals search the optimal value. We take vision of butterfly into the local search phase of the HFBOA. Thus, this search stage of butterfly can be formulated as follows:(8)$$X_{i}^{t+1}=X_{i}^{t}+\beta \times \left(X_{i}^{k}-X_{j}^{t}\right)+\alpha \cdot \epsilon$$
where $X_{j}^{t}$ and $X_{i}^{k}$ are the *j*-th and *k*-th agents, which are chosen randomly from the solution space. Further, ϵ is a random value such that ϵ∈[−0.5,0.5]. α is a random number in [0, 1]. The attractiveness β can be formulated as:(9)$$\beta =\beta_{0}\cdot e^{-R_{ij}}$$
where $\beta_{0}$ is the attractiveness when R=0. The initial value of parameter β is usually set to 1, that is, $\beta_{0}$ = 1. $R_{ij}$ represents the distance between $X_{i}$ and $X_{j}$, which calculates by the 2-norm. The formulation of $R_{ij}$ is:(10)$$R_{ij}=||X_{i}-X_{j}|{|}_{2}$$

According to the four phase analysis, the optimization process of the proposed HFBOA can be shown briefly in Figure 1.

### 3.5. Switch Parameter sp

The switch parameter sp is set to convert the normal global search and the intensive local search. In each iteration, it randomly generates a number in [0, 1], which is compared with the switch probability *sp* to decide whether to conduct a global search and local search. While the value of sp is set to 0; that is, only the local search stage is performed. On the contrary, only the global search phase is carried out with the value sp taken to 1.

### 3.6. Chaotic Map and Parameter α

Chaos is a relatively common phenomenon in nonlinear systems. Logistic mapping [23] is one classical mapping of the one-dimensional maps, it is defined as:(11)$$z_{n+1}=\mu \cdot z_{n}\cdot \left(1-z_{n}\right)$$
where μ denotes the chaotic factor, μ∈(0,4]. The chaotic system is characterized by initial value sensitivity. Therefore, we analyze the speciality of the logistic mapping, where Figure 2 shows chaotic bifurcation and the Lyapunov exponent of the chaos. When μ = 4 and z(0) = 0.35, the chaotic sequence and Lyapunov exponent of logistic mapping are in (0, 1) and 0.6839, respectively.

As we known, as long as the initial value is not 0.25, 0.5, and 0.75, the iterative value will not produce a fixed point of the logistic mapping strategy. Thus, this chaotic mapping is used to update the parameter *c* of the proposed HFBOA, where μ = 4 and $c_{0}$ is set to 0.35 in the following experiments.

The parameter α is updated using chaotic strategy, which substituted the strategy of rand in (0, 1) each iteration. The formula is as follows:(12)$$\alpha_{t+1}=4\alpha_{t}\cdot \left(1-\alpha_{t}\right)$$
where the initial value of parameter α is set to 0.2, and the maximum number of iterations is set to 500 during the optimization process. Thus, the iteration results of $\alpha_{t}$ and $\alpha_{t}^{2}$ are shown in the Figure 3. The reason the initial value α is set to 0.2, not 0.35, is the direct difference of parameters *c* and α.

In this paper, parameters *c* and α are updated by logistic mapping. It can been seen from Figure 2 and Figure 3 that the values of *c* and α are in (0, 1), which is the same as a rand number in (0, 1) using function rand each iteration.

### 3.7. Complexity Analysis

The time complexity is a important factor, which can reflect the performance of the algorithm in a way. It is necessary to compute the time complexity of the algorithm when the algorithm has a finite time to find the global optimal value. The pseudo-code of the proposed algorithm is shown in Algorithm 1, and it shows the steps of the proposed HFBOA.
**Algorithm 1.** The pseudo-code of HFBOA
Generate the initialize population of the butterflies $X_{i}\left(i=1,2,\cdots ,n\right)$ randomly;
Initialize the parameters $\beta_{0},\gamma$, and μ
Define sensory modality *c*, power exponent *a* and switch probability *p*
   for i=1:n
      Calculate the fitness value of each butterflies
   end for
   while $t<T_{max}$
   for i=1:n
      Update the fragrance of current search agent by Equation (Equation 6)
   for j=1:n
   if $fitness_{i}<fitness_{j}$
      Update the attractiveness β and $R_{ij}$ by Equation (Equation 9) and Equation (Equation 10) respectively
   else
      Continue
   end if
% Case1
   if rand<sp
      Update the position using Equation (Equation 7)
   else
      Update the position using Equation (Equation 8)
   end if
% Case2
   if rand<sp
      Update the position using Equation (Equation 3)
   else
      Update the position using Equation (Equation 8)
   end if
      Calculate the fitness value of each butterflies
      Find the best fitness $F_{i}$
   end for
   end for
      Update the value of power exponent *c* and parameter α using Equations (11) and (12)
      t=t+1
   end while
Output the best fitness

According to the pseudo-code, the time complexity of the proposed method can be computed as follows. The initialization phase depends on randomization, which gives n×d random numbers in (0, 1). For the initialization, there are *n* butterflies and the object function is the *d* dimension of the search space; thus, the initialization step costs O(nd). In addition, the maximum evaluation times $\left(T_{max}\right)$ also influence the time complexity. The computational complexity of calculating the fitness of all agents is O(nd). Updating the position in the HFBOA is $O(n^{2}logn)$, the quick sort is $O(n^{2})$, and updating the parameter costs O(nd). Therefore, the final computational complexity of HFBOA is as follows: (13)$$O\left(HFBOA\right)=O\left(nd\right)+O\left(T_{max}\right)O\left(nd+n^{2}logn+n^{2}+nd\right)\approx O\left(T_{max}\right)O\left(nd+n^{2}logn+n^{2}+nd\right)=O\left(T_{max}\times n\times \left(2d+nlogn+n\right)\right)$$

However, the original BOA has the same initialization numbers (*n*), maximum evaluation times ($T_{max}$), and the dimension of search space (*d*) as the proposed HFBOA. The time complexity of the initialization phase is O(nd); calculating the fitness of all agents is $O(T_{max}nd)$; updating the position is $O(T_{max}n^{2}logn)$; the quick-sort is $O(T_{max}n)$; updating the parameter costs $O(T_{max}nd)$. Hence, the time complexity of the BOA is: (14)$$O\left(HFBOA\right)=O\left(nd\right)+O\left(T_{max}nd\right)+O\left(T_{max}n^{2}logn\right)+O\left(T_{max}n\right)+O\left(T_{max}nd\right)=O\left(nd\right)+O\left(T_{max}\times n\times \left(2d+nlogn+1\right)\right)$$

Distinctly, although the complexity of the proposed HFBOA is higher than BOA, the performance of the modified algorithm is superior to the BOA and both of them are in the same order of magnitude. The scalability, convergence accuracy, optimization capability, and robustness of the proposed method can be proved in the following experiments, including the scalability test, 12 benchmark functions test, and constrained engineering problems test, respectively.

## 4. Results of Experiments

In this section, the performance of the HFBOA is substantiated extensively. To verify the performance of the proposed algorithm, 12 benchmark functions from the CEC benchmark functions were tested. Moreover, three experiments were performed with proposed algorithms and other well-known meta-heuristic methods for scalability analysis and statistical analysis, respectively. In addition, the proposed HFBOA was also applied to deal with the six constrained engineering problems.

The experiments were carried out on the same experimental platform. The comparison of all algorithms was conducted in MATLAB 2018a installed over Windows 10 (64 bit), Intel (R) Core (TM) i5-10210U, and @2.11G with 16.0 GB of RAM.

Different types of benchmark functions can help comprehensively evaluate the performance of all competitions in a study of the proposed method. Table 1 shows the 12 benchmark functions, including four unimodal (F1–F4), three multimodal (F5–F7), two fixed (F8, F9) [9], shifted (F10), rotated (F11), and rotated and shifted functions (F12) [24].

The performance of the HFBOA was proved by a set of statistical tests conducted on three criteria, the mean (Mean), standard deviation (Std), and success rate ($S_{r}$); values of all runs are presented. Here, the success rate ($S_{r}$) can be calculated as follows:(15)$$S_{r}=\frac{M_{su}}{M_{all}}\times 100\%$$
where $M_{all}$ denotes the total number of optimization test runs, and $M_{su}$ is the times the algorithm successfully reached the specified value. Here, the specified values are shown in Table 1.

### 4.1. Scalability Analysis of Comparison with Improved Algorithms

The performance of the modified butterfly optimization algorithms was improved to a certain extent. Thus, two test functions F1 and F7 from Table 1 were used to verify the performance of HFBOA compared with four improved algorithms, namely LBOA [16], IBOA [13], MBOA [14], and HPSOBOA [15]. The values of parameter settings of the algorithms were from the original references. In addition, four different dimensions of scalability analysis were considered: 30, 100, 500, and 1000. The same conditions were constructed using 30 individual butterflies with 600 iterations.

It can be seen from Table 2 that the mean optimal value of the test function increased when number of dimension increased. Table 2 shows the comparison results of different comparison algorithms, and the bold face indicates the superior performance of the HFBOA in F1 and F7, except MBOA. Generally speaking, it can be considered as a large-scale complex problem when the dimension of the test function exceeds 300.

### 4.2. Results of Comparison with Meta-Heuristic Algorithms

To verify the performance of the proposed HFBOA, we compared the proposed algorithm with several meta-heuristic algorithms, namely, PSO [1], CS [25], FA [8], GWO [9], HBO [26], and BOA [12]. The parameter settings of the seven approaches are shown in Table 3. The population number of each algorithm was set to 30, and the max iteration was set to 600. Moreover, statistical tests were conducted on three divisions, and each algorithm was run 30 times, independently.

The comparison results of the mean (Mean), standard deviation (Std), and success rate ($S_{r}$) are shown in Table 4. The Wilcoxon rank-sum (WRS) [27] test was used to verify the significance of the proposed method when compared with other algorithms, and the Friedman rank [28] test was used to rank compared approaches. The alpha was set to 0.05 in the WRS and Friedman rank test. Two hypotheses (null and alternative) were used to prove the effectiveness of statistical tests. According to the statistical value, the null was accepted if the statistical value was greater than the value of alpha; otherwise, the alternative was accepted. The *p*-value of the WRS test and the Friedman rank depicted that this supremacy was statistically significant.

Table 4 shows that the HFBOA yielded the best results on the 12 test functions with Dim = 30 except F10 and F12. For F9, the HFBOA obtained the optimal fitness value, which was close to other algorithms, but slightly worse. However, for F5 and F7, the CS algorithm also obtained the best solution of the theoretical optimal value. Additionally, combining the comparison results in Table 4, we see that the HFBOA was better than others in the $S_{r}$ rank, which was set to the specified value. The $S_{r}$ of HFBOA was 100% except F10 and F12 because the complexity of function F10 and F12 was higher than the others in the 12 CEC benchmark functions. According to the Friedman test results, the order of seven comparison algorithms was HFBOA > CS > GWO > BOA > FA > HBO > PSO. Note, the last row in Table 4, the Friedman rank results of the comparison algorithms depicted that the supremacy of the proposed method was statistically significant.

As can be concluded from Table 4, the proposed HFBOA has superior convergence accuracy and optimization capability in unimodal and multimodal functions than other comparison algorithms, especially the basic BOA. For the fixed functions F8 and F9, the optimization results of CS algorithm were slightly better than HFBOA in Std. For the shifted, rotated, and shifted functions F10 and F12, the performance of the proposed method can be further improved in future work.

It can be seen from the convergence curves of comparison algorithms in Figure 4 and Figure 5 that HFBOA has the fastest convergence rate when solving the four unimodal and three multimodal test functions. In functions F1 to F5, and F7, HFBOA obtained the global optimal solution. From Figure 4 and Table 4, HFBOA converged to the global optimal value with a rapid convergence rate in F8, F9, and F11. However, the Std of F8 and F9 by the CS algorithm was better than HFBOA, where NaN means not applicable in Table 4. Furthermore, in function F12, the optimal value of HFBOA was slightly worse than FA from the Mean and Std.

### 4.3. Practical Constrained Engineering Problems

In this subsection, six real-world optimization problems were solved to verify the effectiveness of the HFBOA. The constrained engineering problems (CEPs) are tubular column design [25], three bar truss design [29], tension/compression spring design [25], welded beam design [2], cantilever beam design [29], and speed reducer design [30]. All the considered engineering problems have several inequality constraints that should be handled. They can be formulated by the nonlinear programming (NLP) [31,32], which is formally described as:(16)$$f\left(x\right)=Min\;F\left(x_{i}\right)$$

Subject to: $g_{i}\left(x\right)\leq 0$, i=1,2,⋯,m$lb_{j}\leq x_{j}\leq ub_{j}$, j=1,2,⋯,nwhere $x_{i}={\left(x_{1},x_{2},\cdots ,x_{n}\right)}^{T}\in R^{n}$, f(x) is the objective function, and $g_{i}\left(x\right)\left(i=1,2,\cdots ,m\right)$ is the *i*th inequality constrain, which is defined on $R^{n}$.

The dimension and constrain of the six constrained engineering problems can be summarized in Table 5, and the iteration was set to 300 for each optimization problem using the proposed HFBOA and original BOA. Each constrained engineering problem was optimized as ten times and the statistical results are shown in Table 6.

As can be seen from Table 6, the performance of HFBOA was superior than BOA for solving the constrained engineering problems. The basic BOA had poor stability in CEP3, CEP4, and CEP6 from the statistical results. HFBOA denotes case 1 in the pseudo-code, and HFBOA1 represents case 2 in Algorithm 1, which is used to prove the effect of the updating strategy of parameter α with Logistic mapping instead of function rand. Table 6 shows that HFBOA was superior than HFBOA1 except CEP3 by the statistical results. For CEP3, HFBOA1 was better than HFBOA in the Std, which indicates that the stability of HFBOA1 was higher for solving the tension spring design problem.

The statistical result verified that the performance of the proposed HFBOA was improved, and the robustness of the proposed method was proved for solving different constrained engineering problems. The best results of the state-of-the-art approaches are listed in Table 7, Table 8, Table 9, Table 10, Table 11 and Table 12 in this paper. We compared the best results obtained by the algorithms to show the convergence accuracy and optimization capability of the HFBOA.

#### 4.3.1. Tubular Column Design

The tubular column design [25] is one of the mechanical engineering issues, and can be formulated as follows:

Minimize:$f\left(x_{1},x_{2}\right)=f\left(d,t\right)=9.8x_{1}\cdot x_{2}+2x_{1}$Subject to:$g_{1}=\frac{P}{\pi x_{1}\cdot x_{2}\sigma_{y}}-1\leq 0$$g_{2}=\frac{8PL^{2}}{\pi^{3}Ex_{1}\cdot x_{2}\left(x_{1}^{2}+x_{2}^{2}\right)}-1\leq 0$$g_{3}=\frac{2.0}{x_{1}}-1\leq 0$$g_{4}=\frac{x_{1}}{14}-1\leq 0$$g_{5}=\frac{0.2}{x_{2}}-1\leq 0$$g_{6}=\frac{x_{2}}{0.8}-1\leq 0$where $x_{1}\left(d\right)$ denotes the mean diameter of the column, $x_{2}\left(t\right)$ is the column. Moreover, *P* is a compressive load, $\sigma_{y}$ represents the yield stress, *E* is the modulus of elasticity, ρ is the density, and *L* denotes the length of the designed column.

Variable range:

$2\leq x_{1}\leq 14$ and $0.2\leq x_{2}\leq 0.8$, *P* = 2500 kgf, $\sigma_{y}$ = 500 kgf/cm^2^, *E* = 0.85 × 10${}^{6}$ kgf/cm^2^, *L* = 250 cm, and ρ = 0.0025 kgf/cm^3^.

Table 7 presents the solutions of tubular column design obtained by HFBOA and those reported by CS [25], Rao [33], KH [30], and CSA [33]. As shown, the optimal value of HFBOA was **26.499503**, which means that when $x_{1}$ and $x_{2}$ are set to 5.451157 and 0.291966, respectively, the total cost of the tubular column design is the minimum. It can be concluded that the results obtained by HFBOA were better than those of the previous studies.

#### 4.3.2. Three Bar Truss Design

The mathematical modeling of the three bar truss design [29] is given as follows:

Minimize: $f\left(x_{1},x_{2}\right)=f\left(A_{1},A_{2}\right)=\left(2\sqrt{2A_{1}}+A_{2}\right)\cdot l$Subject to:$g_{1}=\frac{\sqrt{2}\;A_{1}+A_{2}}{\sqrt{2}\;A_{1}^{2}+2A_{1}\;A_{2}\;}\;P-\sigma \leq 0$$g_{2}=\frac{A_{2}}{\sqrt{2}\;A_{1}^{2}+2A_{1}\;A_{2}\;}\;P-\sigma \leq 0$$g_{3}=\frac{1}{A_{1}+\sqrt{2}\;A_{2}\;}\;P-\sigma \leq 0$where *l* is the length of the bar truss, $A_{1}$ and $A_{2}$ denote the cross-sectional areas of the long bar truss and short bar truss, respectively.

Variable range:

$0\leq A_{1},A_{2}\leq 1$, *l* = 100 cm, *P* = 2 kN/cm^2^, and σ = 2 kN/cm^2^.

Table 8 presents the solutions of the three bar truss design obtained by HFBOA and those reported by CS [25], MBA [34], HHO [35], and DSA [36]. As shown, the optimal value of HFBOA was 263.895867, which means that when $x_{1}$ and $x_{2}$ were set to 0.78869137 and 0.408202602, respectively; the total cost of the tension/compression spring was the minimum. The results obtained by HFBOA were better than the CS algorithm and BOA with 300 iterations. However, the results of MBA, HHO, and DSA were slightly better than the proposed method.

#### 4.3.3. Tension/Compression Spring Design

From Ref. [25], the tension/compression spring design was modeled as follows:

Minimize: $f\left(x_{1},x_{2},x_{3}\right)=f\left(d,D,N\right)=x_{1}^{2}\cdot x_{2}x_{3}+2x_{1}^{2}\cdot x_{2}$Subject to:$g_{1}=1-\frac{x_{2}^{3}x_{3}}{71785x_{1}^{4}}\leq 0$$g_{2}=\frac{4x_{1}^{2}-x_{1}x_{2}}{12566(x_{1}\cdot x_{2}^{3}-x_{2}^{4})}+\frac{1}{5108x_{2}^{2}}-1\leq 0$$g_{3}=1-\frac{140.45x_{2}}{x_{1}^{2}x_{3}}\leq 0$$g_{4}=\frac{x_{1}+x_{2}}{1.5}-1\leq 0$

The parameters d,D, and *N* are the three design variables. Where $x_{1}\left(d\right)$ denotes the wire diameter, $x_{2}\left(D\right)$ represents the mean coil diameter, and $x_{3}\left(N\right)$ is the number of active coils.

Variable range:

$0.25\leq x_{1}\leq 1.3$, 0.05≤t≤2.0, and $2\leq x_{3}\leq 15$.

Table 9 presents the solutions of the tension/compression spring obtained by HFBOA and those reported by PSO [2], GWO [9], WOA [37], and GSA [9]. As shown, the optimal value of HFBOA was **0.012666**, which means that when $x_{1}$, $x_{2}$, and $x_{3}$ are set to 0.051841, 0.360377, and 11.078153, respectively, the total cost of the tension/compression spring is the minimum. It can be seen from Table 9 that the results obtained by HFBOA are superior than those of the previous studies, except the GWO algorithm.

#### 4.3.4. Welded Beam Design

There were four main constraints and other side constraints of the welded beam design. τ is the shear stress, δ denotes the bending stress in the beam, $P_{c}$ is the buckling load on the bar, σ is the end deflection of the beam. The mathematical modeling of the welded beam design [2] can be stated as follows:

Minimize: $f\left(x_{1},x_{2},x_{3},x_{4}\right)=f\left(h,l,t,b\right)=1.10471x_{1}^{2}\cdot x_{2}+0.04811x_{3}\cdot x_{4}\left(14.0+x_{2}\right)$Subject to:$g_{1}=\tau_{x}-\tau_{max}\leq 0,$$g_{2}=\sigma_{x}-\sigma_{max}\leq 0,$$g_{3}=\delta_{x}-\delta_{max}\leq 0,$$g_{4}=x_{1}-x_{1}\leq 0,$$g_{5}=P-P_{c}\;\left(x\right)\leq 0,$$g_{6}=0.125-x_{1}\leq 0,$$g_{7}=1.1047x_{1}^{2}+0.04811x_{3}\;x_{4}\;\left(14.0+x_{2}\;\right)-5.0\leq 0.$where $x_{1}\left(h\right)$ is the thickness of the weld, $x_{2}\left(l\right),x_{3}\left(t\right)$, and $x_{4}\left(b\right)$ denote the length of the attached part, the height, and the thickness of the bar, respectively. Additionally,

$\tau \left(x\right)=\sqrt{{\left(\tau^{{}^{\prime }}\right)}^{2}+2\tau^{{}^{\prime }}\tau^{{}^{\prime \prime }}\frac{x_{2}}{2R}+{\left(\tau^{{}^{\prime \prime }}\right)}^{2}}$,

$\tau^{{}^{\prime }}=\frac{P}{\sqrt{2}x_{1}x_{2}}$, $\tau^{{}^{\prime \prime }}=\frac{MR}{J}$, $M=P(L+x_{2}/2)$,

$R=\sqrt{\frac{x_{2}^{2}}{4}+{\left(\frac{x_{1}+x_{3}}{2}\right)}^{2}}$, $J=2\sqrt{2}x_{1}x_{2}\left[\frac{x_{2}^{2}}{4}+{\left(\frac{x_{1}+x_{3}}{2}\right)}^{2}\;\right]$, $\sigma \left(x\right)=\frac{6PL}{x_{4}\;x_{3}^{2}}$, $\delta \left(x\right)=\frac{6PL^{3}}{Ex_{4}x_{3}^{2}}$,

$P_{c}\left(x\right)=\frac{4.013E\sqrt{(x_{3}^{2}\;x_{4}^{6})/36}}{L^{2}}\;\left(1-\frac{x_{3}}{2L}sqrt\frac{E}{4G}\right)$,

*P* = 6000 lb, *L* = 14 in., $\tau_{max}$ = 13,600 psi, $\sigma_{max}$ = 30,000 psi, $\delta_{max}$= 0.25 in., *E* = 30 × 10${}^{6}$ psi, *G* = 12 × 10${}^{6}$ psi

Variable range:

$0.1\leq x_{1}\leq 2$, $0.1\leq x_{2}\leq 10$, $0.1\leq x_{3}\leq 10$ and $0.1\leq x_{4}\leq 2$.

Table 10 presents the solutions obtained by HFBOA and those reported by GSA [9], GWO [9], WOA [37], and DSA [36]. As shown, the optimal value of HFBOA was **1.725080**, which means that when $T_{s}$, $T_{h}$, *R*, and *L* were set to 0.205607, 3.473369, 9.036766, and 0.205730, respectively, the total cost of the welded beam design was the minimum. Thus, it can be concluded that the results obtained by HFBOA were the best in the comparison algorithms.

#### 4.3.5. Cantilever Beam Design

The variables of cantilever beam design were the $x_{1}$ to $x_{5}$ (heights or widths) of the different beam elements and the thickness was held fixed in the problem. The mathematical modeling of the cantilever beam design [25] is given as follows:

Minimize: $f\left(x_{1},x_{2},x_{3},x_{4},x_{5}\right)=0.0624\left(x_{1}+x_{2}+x_{3}+x_{4}+x_{5}\right)$Subject to:$g_{1}=\frac{61}{x_{1}^{3}}+\frac{37}{x_{2}^{3}}+\frac{19}{x_{3}^{3}}+\frac{7}{x_{4}^{3}}+\frac{1}{x_{5}^{3}}-1\leq 0$where $x_{1}$ to $x_{5}$ denote heights or widths of the different beam elements.

Variable range:

$0.01\leq x_{i}\leq 100$ and i=1,2,3,4,5.

Table 11 presents the solutions of the cantilever beam obtained by HFBOA and those reported by CS [25], MMA [29], SOS [38], and MFO [39]. As shown, the optimal value of HFBOA was **1.339963**, which means that when $x_{1}$, $x_{2}$, $x_{3}$, $x_{4}$, and $x_{5}$ were set to 6.016838, 5.313519, 4.495334, 3.495149, and 2.152926, respectively, the total cost of the cantilever beam was minimum. As can seen from Table 11, the results obtained by HFBOA were better than the comparison approaches.

#### 4.3.6. Speed Reducer Design

According to Ref. [30], the optimization problem of speed reducer design can be mathematically formulated as follows:

Minimize: $f\left(x_{1},x_{2},x_{3},x_{4},x_{5},x_{6},x_{7}\right)=0.7854x_{1}x_{2}^{2}\left(3.333x_{3}^{2}+14.9334x_{3}-43.0934\right)-1.508x_{1}\left(x_{6}^{2}+x_{7}^{2}\right)+7.4777\left(x_{6}^{3}+x_{7}^{3}\right)+0.7854\left(x_{4}x_{6}^{2}+x_{5}x_{7}^{2}\right)$Subject to:$g_{1}=\frac{27}{x_{1}x_{2}^{2}x_{3}}-1\leq 0$$g_{2}=\frac{397.5}{x_{1}x_{2}^{2}x_{3}^{3}}-1\leq 0$$g_{3}=\frac{1.93x_{4}^{3}}{x_{2}x_{3}x_{6}^{4}}-1\leq 0$$g_{4}=\frac{1.93x_{5}^{3}}{x_{2}x_{3}x_{6}^{4}}-1\leq 0$$g_{5}=\frac{\sqrt{745{\left(\frac{x_{4}}{x_{2}x_{3}}\right)}^{2}+16.9\times 10^{6}}}{110x_{6}^{3}}-1\leq 0$$g_{6}=\frac{\sqrt{745{\left(\frac{x_{5}}{x_{2}x_{3}}\right)}^{2}+157.5\times 10^{6}}}{85x_{7}^{3}}-1\leq 0$$g_{7}=\frac{x_{2}x_{3}}{40}-1\leq 0$$g_{8}=\frac{5x_{2}}{x_{1}}-1\leq 0$$g_{9}=\frac{x_{1}}{12x_{2}}-1\leq 0$$g_{10}=\frac{1.5x_{6}+1.9}{x_{4}}-1\leq 0$$g_{11}=\frac{1.1x_{7}+1.9}{x_{5}}-1\leq 0$where $x_{1}$ is the face width, $x_{2}$ denotes the module of teeth, $x_{3}$ is the number of the teeth on pinion. $x_{4}$ is the length of shaft 1 between bearings, $x_{5}$ represents the length of shaft 2 between bearings, $x_{6}$ and $x_{7}$ are, respectively, the diameter of shaft 1 and shaft 2.

Variable range:

$2.6\leq x_{1}\leq 3.6$, $0.7\leq x_{2}\leq 0.8$, $17\leq x_{3}\leq 28$, $7.3\leq x_{4}\leq 8.3$, $7.3\leq x_{5}\leq 8.3$, $2.9\leq x_{4}\leq 3.9$ and $5.0\leq x_{4}\leq 5.5$.

Table 12 presents the solutions obtained by HFBOA and those reported by CS [25], KH [30], MFO [39], and DSA [36]. As shown, the optimal value of HFBOA was 2999.0919, which means that when $x_{1}$, $x_{2}$, $x_{3}$, $x_{4}$, $x_{5}$, $x_{6}$, and $x_{7}$ were set to 3.500036, 0.700001, 17, 7.3, 7.800207, 3.458402, and 5.245883, respectively, for speed reducer design. It can be seen from Table 12 that the results obtained by DSA were better than other algorithms. Moreover, the optimal value of HFBOA was slightly worse than the KH algorithm and DSA.

## 5. Discussion

Table 2 shows that the performance of the HFBOA in F1 and F7 was superior to the other improved BOA methods, except MBOA, for solving the high-dimensional optimization problems. For the test results of the 12 benchmark functions in Table 4, the proposed method had better global capability and higher convergence rate. According to the statistical test, the order of seven comparison algorithms was HFBOA > CS > GWO > BOA > FA > HBO > PSO for solving the twelve functions. Moreover, the results on the engineering constrained optimization tasks show that the proposed HFBOA had the vast potential ability to deal with real-world problems as well. However, the performance of the HFBOA should be also enhanced, while the parameters can be reduced without degrading performance.

## 6. Conclusions

To effectively solve constrained engineering problems, a novel hybrid-flash butterfly optimization algorithm (HFBOA) is proposed. The HFBOA combines smell and vision for foraging of global optimization and local optimization, respectively. Besides, updating the control parameters by logistic mapping is synchronously applied into the HFBOA for enhancing the global optimal ability. To evaluate the performance of the HFBOA, experiments were compared with the proposed algorithm and other meta-heuristic algorithms for statistical analysis on 12 benchmark functions.

Compared with seven algorithms, the results of the experiments show that the HFBOA has a significant improvement in terms of solution accuracy performance and convergence speed. Furthermore, the statistical test and complexity analysis are used to verify the efficiency of the HFBOA from different aspects. Moreover, the results on engineering design problems show that HFBOA has the vastly potential ability to deal with real-world problems as well.

In the future work, we will focus on the following tasks:We will theoretically prove the convergence and steady properties of the proposed HFBOA using Markov chain theory [40].Due to the high complexity of the main framework of HFBOA, we will further enhance the HFBOA on the premise of ensuring the optimization precision with Quantum theory.The proposed HFBOA will be further applied to solve the three-dimensional wireless sensor network node deployment problem.

## Figures and Tables

**Figure 1 entropy-24-00525-f001:**
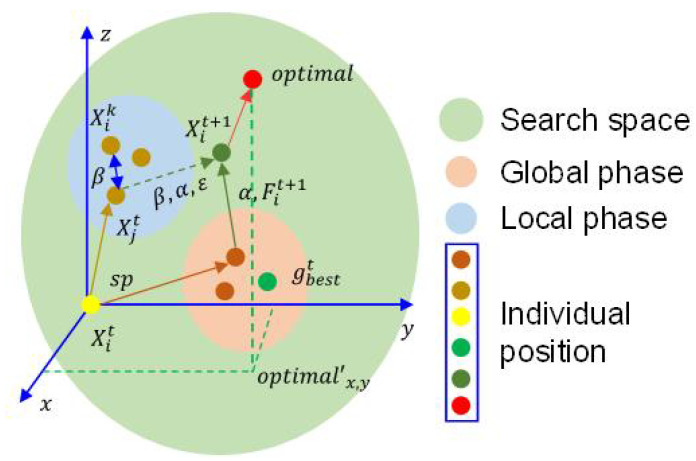
Optimization process of the proposed HFBOA in brief.

**Figure 2 entropy-24-00525-f002:**
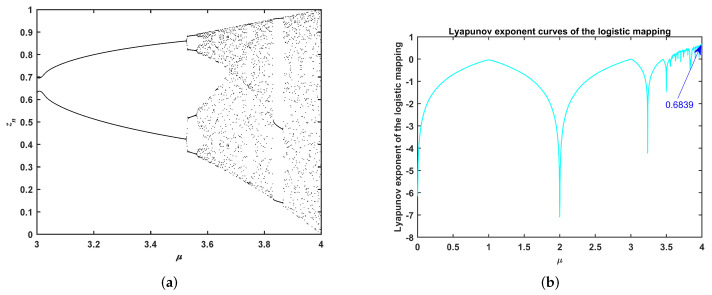
Bifurcation and Lyapunov exponent of the logistic mapping. (**a**) Bifurcation of Logistic mapping. (**b**) Lyapunov exponent of logistic mapping.

**Figure 3 entropy-24-00525-f003:**
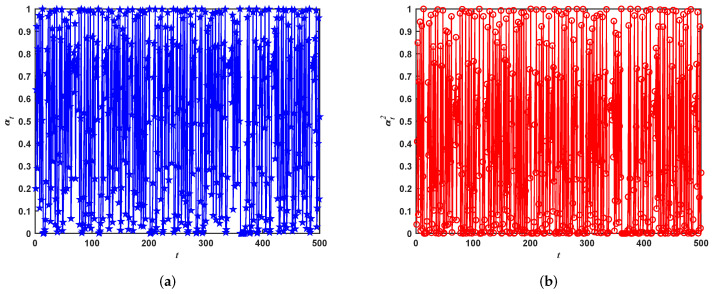
The iterative curves of α and $\alpha^{2}$. (**a**) Iterative curve of parameter α. (**b**) Iterative curve of parameter $\alpha^{2}$.

**Figure 4 entropy-24-00525-f004:**
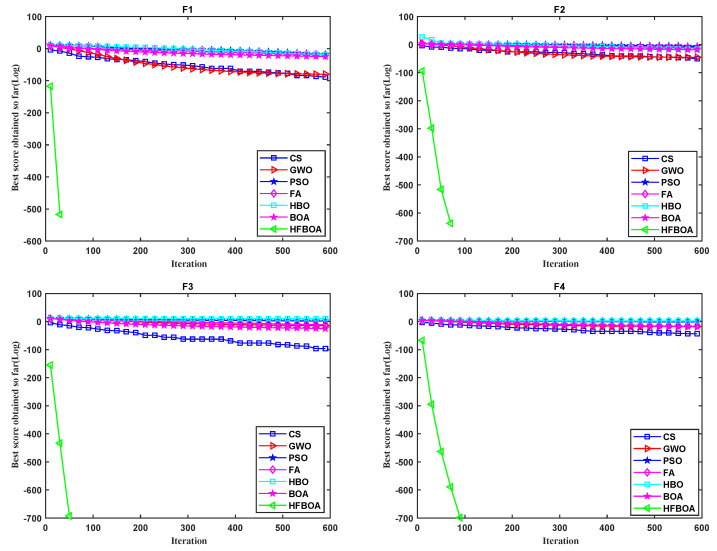
Convergence curves of HFBOA for test functions F1 to F6.

**Figure 5 entropy-24-00525-f005:**
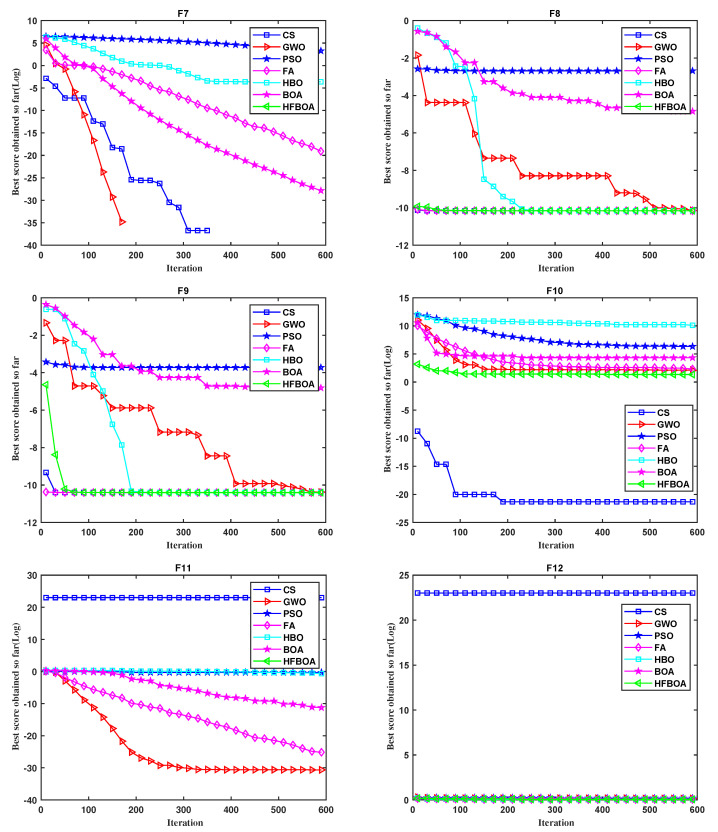
Convergence curves of HFBOA for test functions F7 to F12.

**Table 1 entropy-24-00525-t001:** Benchmark test functions.

Fun	Function Name	Range	Dim	Type	Optimal	Accept
F1	Sphere	[−100, 100]	30	U	0	$10^{-35}$
F2	Schwefel 2.22	[−10, 10]	30	U	0	$10^{-35}$
F3	Schwefel 1.2	[−100, 100]	30	U	0	$10^{-35}$
F4	Schwefel 2.21	[−100, 100]	30	U	0	$10^{-35}$
F5	Rastrigin	[−5.12, 5.12]	30	M	0	$10^{-20}$
F6	Ackley	[−32, 32]	30	M	0	$10^{-15}$
F7	Griewank	[−600, 600]	30	M	0	$10^{-20}$
F8	Shekel 5	[0, 10]	4	fixed	−10.1532	−10.1530
F9	Shekel 7	[0, 10]	4	fixed	−10.4028	−10.4020
F10	Shifted schwefel 1.2	[−100, 100]	30	U	0	$10^{-5}$
F11	Rotated griewank	[−10, 10]	30	M	0	$10^{-5}$
F12	Rotated and shifted ackley	[−32, 32]	30	M	0	$10^{-0}$

**Table 2 entropy-24-00525-t002:** Comparison results of improved algorithms.

Fun	Algorithm	Dim = 30		Dim = 100		Dim = 500		Dim = 1000	
		Mean/$S_{r}$	Std	Mean/$S_{r}$	Std	Mean/$S_{r}$	Std	Mean/$S_{r}$	Std
F1	BOA	1.41 × 10^−11^/0.00	1.25 × 10^−12^	1.60 × 10^−11^/0.00	1.25 × 10^−12^	1.63 × 10^−11^/0.00	1.29 × 10^−12^	1.67 × 10^−11^/0.00	1.33 × 10^−12^
	LBOA	4.82 × 10^−13^/0.00	5.00 × 10^−13^	5.74 × 10^−13^/0.00	6.65 × 10^−13^	7.03 × 10^−13^/0.00	6.61 × 10^−13^	7.35 × 10^−13^/0.00	6.96 × 10^−13^
	IBOA	1.27 × 10^−33^/43.33	1.74 × 10^−33^	3.05 × 10^−33^/23.33	6.57 × 10^−33^	4.74 × 10^−33^/20.00	8.59 × 10^−33^	2.18 × 10^−32^/13.33	6.00 × 10^−32^
	MBOA	**0.00 × 10^0^/100.00**	**0.00 × 10^0^**	**0.00 × 10^0^/100.00**	**0.00 × 10^0^**	**0.00 × 10^0^/100.00**	**0.00 × 10^0^**	**0.00 × 10^0^/100.00**	**0.00 × 10^0^**
	HPSOBOA	6.96 × 10^−46^/100.00	3.58 × 10^−45^	7.89 × 10^−51^/100.00	4.17 × 10^−50^	1.95 × 10^−35^/100.00	6.00 × 10^−35^	8.09 × 10^−42^/100.00	4.26 × 10^−41^
	HFBOA	**0.00 × 10^0^/100.00**	**0.00 × 10^0^**	**0.00 × 10^0^/100.00**	**0.00 × 10^0^**	**0.00 × 10^0^/100.00**	**0.00 × 10^0^**	**0.00 × 10^0^/100.00**	**0.00 × 10^0^**
F7	BOA	9.02 × 10^−13^/0.00	8.90 × 10^−13^	1.35 × 10^−11^/0.00	6.38 × 10^−12^	1.91 × 10^−11^/0.00	1.43 × 10^−12^	1.83 × 10^−11^/0.00	1.74 × 10^−12^
	LBOA	4.38 × 10^−14^/0.00	1.25 × 10^−13^	6.05 × 10^−13^/0.00	9.55 × 10^−13^	8.91 × 10^−13^/0.00	9.60 × 10^−13^	8.24 × 10^−13^/0.00	9.98 × 10^−13^
	IBOA	**0.00 × 10^0^/100.00**	**0.00 × 10^0^**	**0.00 × 10^0^/100.00**	**0.00 × 10^0^**	**0.00 × 10^0^/100.00**	**0.00 × 10^0^**	**0.00 × 10^0^/100.00**	**0.00 × 10^0^**
	MBOA	**0.00 × 10^0^/100.00**	**0.00 × 10^0^**	**0.00 × 10^0^/100.00**	**0.00 × 10^0^**	**0.00 × 10^0^/100.00**	**0.00 × 10^0^**	**0.00 × 10^0^/100.00**	**0.00 × 10^0^**
	HPSOBOA	**0.00 × 10^0^/100.00**	**0.00 × 10^0^**	**0.00 × 10^0^/100.00**	**0.00 × 10^0^**	**0.00 × 10^0^/100.00**	**0.00 × 10^0^**	**0.00 × 10^0^/100.00**	**0.00 × 10^0^**
	HFBOA	**0.00 × 10^0^/100.00**	**0.00 × 10^0^**	**0.00 × 10^0^/100.00**	**0.00 × 10^0^**	**0.00 × 10^0^/100.00**	**0.00 × 10^0^**	**0.00 × 10^0^/100.00**	**0.00 × 10^0^**

**Table 3 entropy-24-00525-t003:** Parameter settings.

Methods	Parameter Settings
PSO	$c_{1}=c_{2}=2,V_{max}=1,V_{min}=-1,\omega \in \left[0.2,0.9\right]$
CS	$\left[C,p_{1},p_{2}\right]$ from corresponding equations
FA	$\beta_{0}=1,\gamma =1$
GWO	$a_{first}=2,a_{final}=0$
HBO	$P_{a}=0.25$
BOA	$a=0.1,p=0.6,c_{0}=0.01$
HFBOA	$a=0.1,p=0.6,\mu =4,\beta_{0}=1,\alpha_{0}=0.2,c_{0}=0.35$

**Table 4 entropy-24-00525-t004:** Comparison results of HFBOA and other optimization algorithms.

Fun	Item	CS	GWO	PSO	HBO	FA	BOA	HFBOA
F1	Mean	5.02 × 10^−39^	6.05 × 10^−34^	9.03 × 10^−7^	1.65 × 10^−9^	1.99 × 10^−9^	1.41 × 10^−11^	**0.00 × 10^0^**
	Std	1.65 × 10^−38^	1.14 × 10^−33^	1.35 × 10^−6^	2.05 × 10^−9^	1.81 × 10^−10^	1.25 × 10^−12^	**0.00 × 10^0^**
	$S_{r}$	**100.00**	13.33	0.00	0.00	0.00	0.00	**100.00**
	*p*-value	1.21 × 10^−12^	1.21 × 10^−12^	1.21 × 10^−12^	1.21 × 10^−12^	1.21 × 10^−12^	1.21 × 10^−12^	-
	Rank	2.0	3.0	7.0	5.2	5.8	4.0	1.0
F2	Mean	3.77 × 10^−20^	2.37 × 10^−20^	2.02 × 10^−3^	3.75 × 10^−7^	1.83 × 10^−5^	5.58 × 10^−9^	**0.00 × 10^0^**
	Std	7.77 × 10^−20^	2.37 × 10^−20^	2.58 × 10^−3^	1.11 × 10^−6^	1.76 × 10^−6^	6.32 × 10^−10^	**0.00 × 10^0^**
	$S_{r}$	0.00	0.00	0.00	0.00	0.00	0.00	**100.00**
	*p*-value	1.21 × 10^−12^	1.21 × 10^−12^	1.21 × 10^−12^	1.21 × 10^−12^	1.21 × 10^−12^	1.21 × 10^−12^	-
	Rank	2.2	2.8	7.0	5.0	6.0	4.0	1.0
F3	Mean	5.48 × 10^−38^	1.98 × 10^−7^	6.41 × 10^0^	2.02 × 10^4^	1.29 × 10^−4^	1.17 × 10^−11^	**0.00 × 10^0^**
	Std	2.07 × 10^−37^	7.35 × 10^−7^	3.40 × 10^0^	7.57 × 10^3^	1.72 × 10^−4^	1.42 × 10^−12^	**0.00 × 10^0^**
	$S_{r}$	**100.00**	0.00	0.00	0.00	0.00	0.00	**100.00**
	*p*-value	1.21 × 10^−12^	1.21 × 10^−12^	1.21 × 10^−12^	1.21 × 10^−12^	1.21 × 10^−12^	1.21 × 10^−12^	-
	Rank	2.0	4.0	6.0	7.0	5.0	3.0	1.0
F4	Mean	1.44 × 10^−19^	2.25 × 10^−8^	2.63 × 10^−1^	1.13 × 10^1^	2.68 × 10^0^	7.54 × 10^−9^	**0.00 × 10^0^**
	Std	3.03 × 10^−19^	1.98 × 10^−8^	8.77 × 10^−2^	4.82 × 10^0^	3.82 × 10^0^	8.50 × 10^−10^	**0.00 × 10^0^**
	$S_{r}$	0.00	0.00	0.00	0.00	0.00	0.00	**100.00**
	*p*-value	1.21 × 10^−12^	1.21 × 10^−12^	1.21 × 10^−12^	1.21 × 10^−12^	1.21 × 10^−12^	1.21 × 10^−12^	-
	Rank	2.0	3.7	5.3	7.0	5.7	3.3	1.0
F5	Mean	**0.00 × 10^0^**	1.39 × 10^0^	5.01 × 10^1^	1.13 × 10^1^	6.11 × 10^1^	5.23 × 10^1^	**0.00 × 10^0^**
	Std	**0.00 × 10^0^**	3.21 × 10^0^	1.44 × 10^1^	2.90 × 10^0^	1.70 × 10^1^	8.55 × 10^1^	**0.00 × 10^0^**
	$S_{r}$	**100.00**	13.33	0.00	0.00	0.00	43.33	**100.00**
	*p*-value	NaN	1.65 × 10^−10^	1.21 × 10^−12^	1.21 × 10^−12^	2.21 × 10^−6^	1.21 × 10^−12^	-
	Rank	1.5	2.7	5.5	4.4	6.4	3.7	3.7
F6	Mean	**8.88 × 10^−16^**	4.27 × 10^−14^	3.29 × 10^−4^	1.77 × 10^−5^	1.04 × 10^−5^	5.38 × 10^−9^	**8.88 × 10^−16^**
	Std	**0.00 × 10^0^**	3.81 × 10^−15^	2.45 × 10^−4^	2.28 × 10^−5^	7.05 × 10^−7^	1.13 × 10^−9^	**0.00 × 10^0^**
	$S_{r}$	**100.00**	0.00	0.00	0.00	0.00	0.00	**100.00**
	*p*-value	NaN	7.17 × 10^−13^	1.21 × 10^−12^	1.21 × 10^−12^	1.21 × 10^−12^	1.21 × 10^−12^	-
	Rank	1.5	3.0	7.0	5.5	5.5	4.0	1.5
F7	Mean	**0.00 × 10^0^**	3.54 × 10^−3^	2.03 × 10^1^	1.40 × 10^−3^	3.20 × 10^−3^	9.02 × 10^−13^	**0.00 × 10^0^**
	Std	**0.00 × 10^0^**	7.24 × 10^−3^	5.88 × 10^0^	3.77 × 10^−3^	5.22 × 10^−3^	8.90 × 10^−13^	**0.00 × 10^0^**
	$S_{r}$	**100.00**	0.00	0.00	0.00	0.00	0.00	**100.00**
	*p*-value	NaN	5.58 × 10^−3^	1.21 × 10^−12^	1.21 × 10^−12^	1.21 × 10^−12^	1.21 × 10^−12^	-
	Rank	1.9	2.9	7.0	5.2	5.4	3.8	1.9
F8	Mean	**−1.02 × 10^1^**	−9.65 × 10^0^	−5.73 × 10^0^	−9.89 × 10^0^	−9.40 × 10^0^	−4.94 × 10^0^	**−1.02 × 10^1^**
	Std	**3.23 × 10^−15^**	1.54 × 10^0^	3.51 × 10^0^	1.30 × 10^0^	1.99 × 10^0^	7.68 × 10^−1^	4.19 × 10^−6^
	$S_{r}$	**100.00**	6.67	36.67	93.33	86.67	0.00	**100.00**
	*p*-value	5.89 × 10^−2^	3.02 × 10^−11^	7.57 × 10^−2^	1.88 × 10^−9^	1.09 × 10^−6^	3.02 × 10^−11^	-
	Rank	3.5	5.3	4.9	1.4	2.7	6.4	3.8
F9	Mean	**−1.04 × 10^1^**	−1.04 × 10^1^	−7.42 × 10^0^	−1.04 × 10^1^	−9.80 × 10^0^	−4.72 × 10^0^	**−1.04 × 10^1^**
	Std	**8.30 × 10^−15^**	6.66 × 10^−4^	3.73 × 10^0^	7.19 × 10^−2^	1.89 × 10^0^	6.47 × 10^−1^	1.69 × 10^−6^
	$S_{r}$	**100.00**	50.00	60.00	93.33	90.00	0.00	**100.00**
	*p*-value	1.21 × 10^−12^	3.02 × 10^−11^	1.79 × 10^−1^	1.24 × 10^−9^	1.05 × 10^−7^	3.02 × 10^−11^	-
	Rank	4.4	5.4	3.7	1.5	2.9	6.6	3.4
F10	Mean	**2.05 × 10^−10^**	9.22 × 10^0^	6.26 × 10^2^	1.97 × 10^4^	1.08 × 10^1^	4.95 × 10^1^	3.81 × 10^0^
	Std	**4.02 × 10^−10^**	3.60 × 10^0^	2.85 × 10^2^	7.20 × 10^3^	4.60 × 10^0^	2.78 × 10^1^	9.02 × 10^−1^
	$S_{r}$	**100.00**	0.00	0.00	0.00	0.00	0.00	0.00
	*p*-value	3.02 × 10^−11^	6.70 × 10^−11^	3.02 × 10^−11^	3.02 × 10^−11^	2.61 × 10^−10^	3.02 × 10^−11^	-
	Rank	1.0	3.1	6.0	7.0	3.9	5.0	2.0
F11	Mean	1.00 × 10^10^	8.93 × 10^−12^	5.14 × 10^−1^	2.20 × 10^−1^	5.98 × 10^−12^	1.29 × 10^−5^	**0.00 × 10^0^**
	Std	0.00 × 10^0^	3.18 × 10^−11^	3.77 × 10^−1^	1.26 × 10^−1^	1.41 × 10^−12^	5.81 × 10^−6^	**0.00 × 10^0^**
	$S_{r}$	0.00	**100.00**	0.00	0.00	**100.00**	**100.00**	**100.00**
	*p*-value	1.69 × 10^−14^	1.21 × 10^−12^	1.21 × 10^−12^	1.21 × 10^−12^	1.21 × 10^−12^	1.21 × 10^−12^	-
	Rank	7.0	2.1	5.7	5.3	2.9	4.0	1.0
F12	Mean	1.00 × 10^10^	1.13 × 10^0^	1.18 × 10^0^	1.08 × 10^0^	**9.75 × 10^−1^**	1.13 × 10^0^	1.00 × 10^0^
	Std	0.00 × 10^0^	4.64 × 10^−2^	7.25 × 10^−2^	5.57 × 10^−2^	**3.79 × 10^−2^**	4.36 × 10^−2^	5.06 × 10^−2^
	$S_{r}$	0.00	0.00	0.00	0.00	76.67	0.00	43.33
	*p*-value	1.21 × 10^−12^	2.61 × 10^−10^	1.46 × 10^−10^	5.86 × 10^−6^	1.38 × 10^−2^	3.16 × 10^−10^	-
	Rank	7.0	4.7	6.0	3.0	1.1	4.3	1.9
Overall	Avg.rank	3.00	3.56	5.92	4.79	4.44	4.34	**1.94**
	Total rank	2	3	7	6	5	4	**1**

**Table 5 entropy-24-00525-t005:** Parameter of HFBOA for solving the CEPs.

Item	Problems	Dim	Cons	Iter
CEP1	Tubular column design	2	6	300
CEP2	Three bar truss design	2	3	300
CEP3	Tension spring design	3	4	300
CEP4	Welded beam design	4	7	300
CEP5	Cantilever beam design	5	1	300
CEP6	Speed reducer design	7	11	300

**Table 6 entropy-24-00525-t006:** Statistical results of the six CEPs.

Problems	Algorithms	Best	Mean	Std
CEP1	BOA	26.512782	26.611700	6.31 × 10^−2^
	HFBOA	26.499503	26.499571	4.12 × 10^−5^
	HFBOA1	26.499543	26.499662	1.08 × 10^−4^
CEP2	BOA	263.935051	264.254896	1.84 × 10^−1^
	HFBOA	263.895867	263.895929	3.49 × 10^−5^
	HFBOA1	263.895895	263.895993	7.93 × 10^−5^
CEP3	BOA	0.012790	3.6498 × 10^11^	7.81 × 10^11^
	HFBOA	0.012666	0.012781	2.25 × 10^−4^
	HFBOA1	0.012667	0.012711	5.23 × 10^−5^
CEP4	BOA	2.189107	2.1944 × 10^−7^	6.94 × 10^7^
	HFBOA	1.725080	1.725458	3.11 × 10^−4^
	HFBOA1	1.725997	1.727217	1.24 × 10^−3^
CEP5	BOA	1.359825	1.371087	9.59 × 10^−3^
	HFBOA	1.339963	1.339977	7.52 × 10^−6^
	HFBOA1	1.340032	1.340069	3.36 × 10^−5^
CEP6	BOA	3178.596571	2.2771 × 10^1^	3.26 × 10^11^
	HFBOA	2999.091940	2999.129526	4.38 × 10^−2^
	HFBOA1	2999.122912	2999.174810	4.93 × 10^−2^

**Table 7 entropy-24-00525-t007:** Best results of tubular column design.

Item	$x_{1}$	$x_{2}$	$f_{min}$
CS	5.45139	0.29196	26.53217
Rao	5.44	0.293	26.5323
KH	5.451278	0.291957	26.5314
CSA	5.451163397	0.291965509	26.531364472
BOA	5.448426	0.292463	26.512782
HFBOA	5.451157	0.291966	**26.499503**

**Table 8 entropy-24-00525-t008:** Best results of three bar truss design.

Item	$x_{1}$	$x_{2}$	$f_{min}$
CS	0.78867	0.40902	263.9716
MBA	0.788565	0.4085597	263.8958522
HHO	0.788662816	0.4082831338329	**263.8958434**
DSA	0.788675136	0.408248285	**263.8958434**
BOA	0.783880758	0.422200913	263.935051
HFBOA	0.78869137	0.408202602	263.895867

**Table 9 entropy-24-00525-t009:** Best results of tension/compression spring.

Item	$x_{1}$	$x_{2}$	$x_{3}$	$f_{min}$
PSO	0.015728	0.357644	11.244543	0.0126747
GWO	0.05169	0.356737	11.28885	**0.012666**
WOA	0.051207	0.345215	12.004032	0.0126763
GSA	0.050276	0.323680	13.525410	0.0127022
BOA	0.051129	0.341493	12.326899	0.012789
HFBOA	0.051841	0.360377	11.078153	**0.012666**

**Table 10 entropy-24-00525-t010:** Best results of welded beam design.

Item	$x_{1}$	$x_{2}$	$x_{3}$	$x_{4}$	Optimal
GSA	0.182129	3.856979	10.000000	0.202376	1.879952
GWO	0.205676	3.478377	9.036810	0.205778	1.726240
WOA	0.205396	3.484293	9.037426	0.206276	1.730499
DSA	0.205731	3.475599	9.036601	0.205731	1.725555
BOA	0.175591	5.214398	7.785997	0.279475	2.189107
HFBOA	0.205607	3.473369	9.036766	0.205730	**1.725080**

**Table 11 entropy-24-00525-t011:** Best results of cantilever beam design.

Item	CS	MMA	SOS	MFO	BOA	HFBOA
$x_{1}$	6.0089	6.0100	6.01878	5.984871	5.785193	6.016838
$x_{2}$	5.3049	5.3000	5.30344	5.316726	4.942404	5.313519
$x_{3}$	4.5023	4.4900	4.49587	4.497332	4.786671	4.495334
$x_{4}$	3.5077	3.4900	3.49896	3.513616	3.692129	3.495149
$x_{5}$	2.1504	2.1500	2.15564	2.161620	2.585670	2.152926
$f_{min}$	1.33999	1.3400	1.33996	1.339988	1.359825	**1.339963**

**Table 12 entropy-24-00525-t012:** Best results of speed reducer design.

Item	CS	KH	MFO	DSA	BOA	HFBOA
$x_{1}$	3.5015	3.499966	3.507524	3.500006	3.6	3.500036
$x_{2}$	0.7000	0.7	0.7	0.7	0.7	0.700001
$x_{3}$	17.0000	17.00001	17	17	17	17
$x_{4}$	7.6050	7.36601	7.302397	7.300490	7.3	7.3
$x_{5}$	7.8181	7.822665	7.802364	7.8	7.8	7.800207
$x_{6}$	3.3520	3.350358	3.323541	3.350216	3.459341	3.458402
$x_{7}$	5.2875	5.286674	5.287524	5.286759	5.461176	5.245883
$f_{min}$	3000.981	2997.447	3009.571	2996.4034	3178.5965	2999.0919

## Data Availability

Not applicable.

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
