# Peer review of "Hybrid-Flash Butterfly Optimization Algorithm with Logistic Mapping for Solving the Engineering Constrained Optimization Problems"

_entropy, 2022, doi:10.3390/e24040525_

Round 1

Reviewer 1 Report

  1. Explain the significance of the word flash is the name of the proposed algorithm in this study as vision is only added with smell.
  2. What are the variants which are compared as mentioned in lines 54 and 55 and how HFBOA is superior to others.
  3. How the iteration is set to 300? if done empirically then mention the detail.
  4. Variable/ parameters detail of the six practical constrained engineering problems is to be given in respective problems.
  5.  Minor corrections: In line 54, the is to be deleted

In line 165 is to be replaced with in.

In line 218 covers is to be corrected as curves

In line 219 HABOA is to be corrected as HFBOA

Title of the table 6 is to be corrected

In Line 364, Table 14 is to be corrected as Table 12

In Line 376 which is to be replaced with while

Author Response

We appreciate the reviewer for his/her precious and valuable comments that help us improve the quality of the paper. In the revised paper and the response letter, the mentioned issues have been fully considered and addressed.

Comments and Suggestions for Authors

  1. Explain the significance of the word flash is the name of the proposed algorithm in this study as vision is only added with smell.

Response:

The reasons that we use the term “flash” are explained as follows. The “butterfly optimization algorithm” represents the inspiration of the smell perception of butterflies. The “flash” in the title denotes the inspiration of the vision butterfly species and we use it to show that the vision perception considered in the modified algorithm. If we use the “vision” instead, ambiguity will arise. In a word, the term “flash” indicates the novelty of the proposed algorithm and avoids the ambiguity compared to “vision”.

  1. What are the variants which are compared as mentioned in lines 54 and 55 and how HFBOA is superior to others.

Response:

In lines 55-56 of the revised manuscript, we have listed several variants of the original BOA.

  1. How the iteration is set to 300? if done empirically then mention the detail.

Response:

 By a series of experiments, we find that the results of 300 iterations tend to be stable when using the improved algorithm to solve engineering optimization problems. Hence, we set the number of iterations to be 300 in this paper.

  1. Variable/ parameters detail of the six practical constrained engineering problems is to be given in respective problems.

Response:

In lines 262-264, 281-282, 300-301, 311-313, 325-326, 348, 375-377 of the revised manuscript, we have added the parameter’s detail of the six practical constrained engineering problems, also stated as follows.

  • Tubular column design problem

P is a compressive load, x1 (d) denotes the mean diameter of the column, x2 (t) is the columns with thickness,  represents the yield stress, E is a modulus of elasticity, ρ is a density, and L denotes the length of the designed column.

  • Three-bar truss design problem

where l is the length of the bar truss, A1 and A2 denote the cross-sectional areas of the long bar truss and short bar truss, respectively.

  • Tension/compression string design problem

The parameters d, D and N are the three design variables. Where x1 (d) denotes the wire diameter, x2 (D) represents the mean coil diameter, and x3 (N) is the number of active coils.

  • Welded beam design problem

There are four main constraints and other side constraints. τ is the shear stress, σ denotes the bending stress in the beam, Pc is the buckling load on the bar, δ is the end deflection of the beam. This issue has four variables, x1(h) is the thickness of weld, x2(l), x3(t) and x4(b) denote the length of attached part, the height and the thickness of the bar, respectively.

  • Cantilever beam

The design variables of cantilever beam design are the x1 to x5 (heights or widths) of the different beam elements, and the thickness is held fixed in the problem.

  • Speed reducer design problem

This problem has seven variables. x1 is the face width, x2 denotes the module of teeth, x3 is the number of teeth on pinion. x4 is length of shaft 1 between bearings, x5 represents the length of shaft 2 between bearings, x6 and x7 are respectively the diameter of shaft 1 and shaft.

  1. Minor corrections:
  • In line 54, the is to be deleted In line 165 is to be replaced with in.
  • In line 218 covers is to be corrected as curves
  • In line 219 HABOA is to be corrected as HFBOA
  • Title of the table 6 is to be corrected
  • In Line 364, Table 14 is to be corrected as Table 12
  • In Line 376 which is to be replaced with while

Response:

As suggested, we have corrected the mentioned typos and carefully checked the rest of the revised manuscript.

Thanks a lot for the invaluable comments and suggestions. The comments from the editors and reviewers are significant, informative, and constructive, which greatly help us improve the quality of the manuscript.

Reviewer 2 Report

The article presents a hybrid-flash butterfly optimization algorithm for optimizing engineering problems.

It is well written and the proposed algorithm was extensively simulated and compared to other methods.

I have some minor comments:

  1. In the introduction, some of the important meta-heuristics were not mentioned, i.e, ant colony optimization, differential evolution, and heterogeneous distributed bees algorithm.
  2. It is not very clear why the algorithm HFBOA is needed apart from improving the optimization accuracy of the original BOA. What was the gap in the previous approach?
  3. In the parameter settings it is unclear why the parameters 'c' and 'w' for the PSO algorithm were set this way. This algorithm was extensively studied in the last 25 years and the most acceptable parameters are c=1.49618 and w=0.729844.
  4.  It would be beneficial if the authors state the relative improvement if the performance in percentage to fully appreciate the results.

Author Response

Thanks a lot for the invaluable comments and suggestions, these suggestions are very valuable for our paper, and we carefully revise the manuscript according to these suggestions.

Comments and Suggestions for Authors

The article presents a hybrid-flash butterfly optimization algorithm for optimizing engineering problems. It is well written and the proposed algorithm was extensively simulated and compared to other methods.

Response:

Thanks a lot for the invaluable comments. In the revised manuscript, we have further improved the quality based on the given suggestions.

I have some minor comments:

  1. In the introduction, some of the important meta-heuristics were not mentioned, i.e, ant colony optimization, differential evolution, and heterogeneous distributed bees algorithm.

Response:

As suggested, we have added and reviewed the references of important meta-heuristics in lines 23-26 of the revised manuscript.

  1. It is not very clear why the algorithm HFBOA is needed apart from improving the optimization accuracy of the original BOA. What was the gap in the previous approach?

Response:

First, the specificity of some species of butterflies with light sensitivity in nature is not considered in the original BOA design, so the original BOA is not completely inspired by butterflies. We take the vision perception into consideration to improve the algorithm where the local search adopts this strategy. Second, the update strategy of the parameter c is replaced by the logistic mapping, which not only verifies the effectiveness of chaos strategy, but also changes the value range of parameter c in the original BOA. The interval of logistic mapping is (0, 1), while the value range of parameter c of the basic BOA is (0.1, 0.3) from the literature.

  1. In the parameter settings it is unclear why the parameters 'c' and 'w' for the PSO algorithm were set this way. This algorithm was extensively studied in the last 25 years and the most acceptable parameters are c=1.49618 and w=0.729844.

Response:

The parameter setting is a hot point in the early study of PSO algorithm. In [2], a modified PSO algorithm was proposed and there is no parameter “w” in the first type of the PSO algorithms. After that, the parameter setting of the variants of the PSO algorithm becomes more popular by changing the parameters “w” and “c”. Additionally, the parameters “w” and “c” in basic PSO algorithm were set to be “2” in [1]. The parameters “c1” and “c2” were set to be “2” in the formula in the following researches. In the proposed algorithm, we set the values of the parameters by referring to the cited literature. The reason that parameter “w” was set to (0.9, 0.2) dynamically in our work is that we have tested and compared with the interval (0.9, 0.4) [3] in our early work. In a word, there are no “True” or “False” in the parameter setting research of PSO algorithms. The literature mentioned above is listed as follows.

  • Kennedy and R. Eberhart, "Particle swarm optimization," International Conference on Neural Networks, 1995, vol. 4, pp. 1942-1948.
  • Shi and R. Eberhart, "A modified particle swarm optimizer," IEEE International Conference on Evolutionary Computation Proceedings, 1998, pp. 69-73.
  • He, and L. Wang, "An effective co-evolutionary particle swarm optimization for constrained engineering design problems," Engineering Applications of Artificial Intelligence, 2007, vol. 20, no. 1, pp. 89-99.
  1. It would be beneficial if the authors state the relative improvement if the performance in percentage to fully appreciate the results.

***************************************************************************

Response:

As suggested, we have added the comparison for the success rate (Sr) between the improved algorithm with several existing algorithms in Table 2 of the revised manuscript.

Thanks a lot for the invaluable comments and suggestions. The comments from the editors and reviewers are significant, informative, and constructive, which greatly help us improve the quality of the manuscript.
